# Effect of status disclosure on quality of life among people living with HIV/AIDS in Ghana: A health facility-based cross-sectional study

**Farrukh Ishaque Saah**[1,2]*, **Kizito Aidam**[3], Abdul-Aziz Seidu[1,4], Hubert Amu[5], Luchuo Engelbert Bain[6]

**1** Department of Population and Health, Faculty of Social Sciences, University of Cape Coast, Cape Coast, Ghana, **2** Public Health Emergency Operations Centre, Clinical and Public Health Services Department, Ministry of Health, Kigali, Rwanda, **3** Department of Family and Community Health, School of Public Health, University of Health and Allied Sciences, Hohoe, Ghana, **4** College of Public Health, Medical and Veterinary Sciences, James Cook University, Townsville, Australia, **5** Department of Population and Behavioural Sciences, School of Public Health, University of Health and Allied Sciences, Hohoe, Ghana, **6** International Programs, African Population and Health Research Center, Nairobi, Kenya

* fsaahpnur14@uhas.edu.gh

**Data Availability Statement:** The data has been made available as a supporting file.

**Funding:** The authors received no specific funding for this work.

## Abstract

Status disclosure to family and friends among people living with HIV/AIDS (PLWHA) is necessary to ensure social and material support, which are important predictors of quality of life. We examined HIV/AIDS status disclosure and its influence on quality of life (QoL) among PLWHA in a Ghanaian population. This was a health facility-based cross-sectional study of 124 PLWHA accessing antiretroviral therapy at a hospital in the Ashanti Region of Ghana. Data were collected using a pretested questionnaire adopting the WHO HIV-QOL BREF. The data were analysed descriptively and inferentially using STATA 15.0. Statistical significance was set at a p value<0.05 at a 95% confidence interval. Forty-two percent of the respondents disclosed their HIV status. Social support was available for 76.9% of those who had disclosed their status. The mean QoL was highest within the physical domain (14.3 ±2.9), while the psychological domain indicated the lowest quality of life (11.03±3.26). Approximately 47.6% attained the mean overall QoL score (73.18). Being a male predicted lower QoL in all domains than being a female. Being a Muslim, having tertiary education, and being an urban resident significantly predicted better QoL than being a Christian, having no formal education, and being a rural resident, respectively. Respondents' age negatively predicted overall quality of life. Although disclosing one's HIV/AIDS status positively predicted QoL, the difference was not statistically significant. A significantly poor quality of life among patients can drag efforts by Ghana towards achieving the Sustainable Development Goal of reducing the HIV/AIDS pandemic and its associated mortality. Stakeholders in AIDS prevention and management in Ghana should employ innovative methods such as peer support to encourage status disclosure and provide education on the provision of social and emotional support for PLWHA.

**Competing interests:** The authors have declared that no competing interests exist.

## Introduction

The advent of antiretroviral therapy (ART) and effective treatment protocols have provided opportunities for PLWHA to live longer lives; hence, HIV is now perceived as a manageable chronic condition instead of a terminal illness [1]. Indeed, there is conclusive evidence that effective ART reduces the risk of HIV transmission to almost zero when the viral load remains persistently below 1000 copies per ml [2]. The treatment also significantly improves life expectancy and health-related quality of life [3]. The U.S. President's Emergency Plan for AIDS Relief (PEPFAR) recently estimated that approximately 39 million people are living with HIV/AIDS worldwide, and 29.8 million out of this number have access to antiretroviral treatment [4]. Adopting and scaling up the most effective strategies to prevent new infections has been central to the HIV/AIDS control strategy [5]. This strategy is aimed at improving the quality of life (QoL) of people battling the infection [5].

Self-disclosure of sensitive information is beneficial for people's health: lower stress and better psychological well-being [6, 7]. Studies also indicate that individuals who disclosed their results have greater adherence to ART treatments [8–10]. Moreover, disclosure may increase opportunities to receive social support, which may help individuals cope and recover from physical illness and lessen depressive symptoms due to HIV-related physical symptoms [11, 12].

Folasire and colleagues argue that the availability of care and social support from family members and close friends are major determinants of the QoL of PLWHA [13]. Although willing to access this support and care, most PLWHA are concerned about the possibility that disclosing their HIV status to family members and close friends will result in some stigmatization and discrimination, likely affecting their loved ones and personal relationships [14, 15]. The negative attitude of society and HIV-related stigma dissuade PLWHA from disclosing their status, especially in low- and middle-income countries [16, 17]. Stigma has profound implications for HIV prevention, treatment, care, and support [18, 19].

Studies have shown reluctance to disclose HIV status among many sub-Saharan Africans even when they had gone for Voluntary Counselling and Testing, with the few who choose to disclose it being selective in choosing their audience [20–23]. Stigma and discrimination against PLWHA are pervasive and represent significant obstacles to HIV/AIDS prevention efforts, including HIV testing and status disclosure [24, 25]. Internalised and experienced/expected HIV-related stigma, such as guilt, shame, fear of abuse, and rejection, among other things, is a major obstacle to HIV status disclosure in Ghana and some parts of Sub-Saharan Africa [26–28]. In Ghana, HIV status disclosure remains a difficult task for most HIV-positive individuals poses a real challenge in providing treatment and clinical care to PLWHA [29]. Internalised HIV stigma is common among PLWHA in Ghana while community-related stigma has been found to affect HIV status disclosure [30]. Acquiring HIV/AIDS in Ghana is generally considered the result of promiscuity or sexual immorality, so infected individuals are highly discriminated against and stigmatized [31]. PLWHA are subjected to a variety of types of stigmatisations and discrimination, such as rejection from their families and communities, inability to share a meal or a bed, finger-pointing, and exclusion [25, 32]. These discriminations and stigmatization lead to decreased self-esteem, lowered body image, negative feelings about self, depression, anxiety, self-isolation, feelings of guilt and failure to disclose status [33].

Chronic illness has always constituted a barrier to QoL [34, 35]. HIV/AIDS is notorious for hindering the QoL of people [34]. PLWHA face psychological isolation and condemnation from their family, friends, and society because people around them are aware of their HIV status, affecting their physical health even though ARTs help with regaining their strength [36, 37].

Disclosure to family members and friends is indeed necessary to ensure social and/or material support [38], especially when the availability of care and social support from family members and close friends tend to be the major determinants of the QoL of those living with HIV in resource-limited settings [13]. It is also noted that the desire for care and support enforces the willingness to disclose one's HIV status; however, this is withheld where support is not expected [13].

There is a paucity of information regarding disclosure and QoL among PLWHA in the Obuasi East District of the Ashanti Region. It is therefore of prime importance to know what the effect of HIV status disclosure has been on PLWHA. Therefore, we examined the influence of HIV/AIDS status disclosure on quality of life among PLWHA in the Obuasi East District of the Ashanti Region of Ghana.

## Methods and materials

### Ethics statement

Ethical approval was obtained from the Research Committee of the District Health Directorate (OMHD/2020/24). Administrative permission was sought from the hospital management. Prior to inclusion in the study, written informed consent was obtained from patients, which involved signing/thumbprinting an information sheet describing and explaining the purpose of the research, benefits, potential harms and confidentiality of respondents to prevent victimization of any form. All research was performed in accordance with relevant guidelines/regulations. The study did not collect any personal identifying information of the respondents; instead, pseudonyms were used to protect the anonymity and privacy of the respondents. The data have been deidentified and stored on the password-protected computer of the authors without access by any third parties to ensure the confidentiality of the respondents.

### Study design

This study adopted a health facility-based cross-sectional design. It involved the collection of quantitative data from a section of PLWHA from a health facility at a single point in time. This report followed the Strengthening the Reporting of Observational Studies in Epidemiology (STROBE) guidelines.

### Study site and population

The study was carried out at a hospital in the Obuasi East District in the Ashanti Region of Ghana. The district was carved out of the Obuasi Municipality to become one of the 260 districts in Ghana [39]. The capital is Tutuka. The hospital served as one of the three designated study sites for a planned intervention study, representing a pivotal hub for healthcare services in the region. Notably, it was among the earliest facilities to initiate ART services within the district, signifying its crucial role in providing HIV/AIDS treatment and care. As a primary centre for ART services, the hospital plays a vital role in addressing the healthcare needs of individuals living with HIV/AIDS in the local community. This study site was selected due to its significance in delivering comprehensive HIV/AIDS care and its accessibility to a diverse patient population within the district.

The study involved PLWHA who were accessing ART services at a health facility in the Obuasi Municipality. This population was chosen because they are experiencing service that is meant to impact the phenomena of this study, the decision to disclose HIV serostatus and QoL. However, PLWHA who were seriously ill were excluded. This included individuals experiencing a high burden of symptoms related to HIV/AIDS, such as pain, fatigue, or

cognitive impairment, and those with significant limitations in their ability to perform daily tasks due to HIV/AIDS-related symptoms or complications.

## Sampling

The hospital's ART clinic has 210 PLWHA registrants who are currently receiving ART services. Hence, the study utilized the sample size calculation formula by Yamane [40] $n = \frac{N}{1+N(e)^2}$, where N is the population size (210) and $e$ is the margin of error at a 95% confidence interval (0.05). The estimated sample size was 138.

The study adopted a simple random sampling technique, a balloting approach. The ART Register was used as the sampling frame, and balloting was performed on the registration numbers. Individuals whose numbers were selected and consented were included in the study. However, where a selected patient refused to consent, they were replaced with other patients from the register who provided consent.

## Study variables

The dependent variable in this study was health-related QoL among PLWHA. This variable was determined using the WHOQOL BREF tool [41]. The independent variables included the sociodemographic characteristics of the respondents, such as age, sex, religion, ethnicity, level of education, occupation, duration of infection awareness, and HIV serostatus disclosure. HIV serostatus disclosure in this study referred to disclosure of one's HIV/AIDS status to a third party (e.g family, friends, etc.) from a healthcare service provider.

## Procedures

Data were collected using a pretested questionnaire (Cronbach's alpha reliability = 0.804). The questionnaire was sectioned into three parts: sociodemographic information, status disclosure, and quality of life. The WHOQOL-HIV BREF instrument was adopted to assess the respondents' quality of life. The instrument is used to appraise QoL as well as respondents' overall perception of their health using 31 questions evaluating QoL from six domains, namely, physical, psychological, level of independence, social relation, environmental, and spiritual domains. These domains are scored on a 5-point Likert scale with 5-very good and 1-very poor [41]. Persons accessing ART at the hospital were selected and approached after they had received their service, and the purpose of the study was explained to them. Upon completion of the study participation process and assurance of ethical concerns and all other concerns addressed, a person living with HIV who consented to participate in the study was interviewed. Data were collected with the help of three research assistants between October 15 and November 10, 2020. Respondents who could not read or understand the questionnaire were assisted, and the questions were explained to them.

## Data analyses

Data (S1 Data) collated were coded and entered into the EpiData version 4.3 template and exported to Stata version 15 (Stata Corporation, College Station, TX, USA) for cleaning and analysis. Descriptive statistics such as frequency, mean, and percentage were generated to summarize the variables. A multiple linear regression analysis was conducted to test the predictors of QoL. QoL is normally distributed. The data was tested to confirm the key assumptions for conducting a linear regression test. Categorical independent variables were converted into dummy variables in preparation to set up the model. The linearity assumption was established to confirm linear relationship between the variables. All the independent variables except

status disclosure were tested for their strength of association and prediction in the initial step. A second multiple linear regression was conducted with all the independent variables inclusive of status disclosure to determine the effects of status disclosure on QoL and its predictors. Statistical significance was considered at a p value of less than 0.05 at a 95% confidence interval. The QoL variable was analysed following the WHO guidelines for analysis of the WHOQOL-HIV BREF tool and grouped into six intermediary variables: physical, psychological, social, environmental, level of independence, and spiritual domains [41]. Total scores of each of the six domains were calculated from the sum of the responses of the domain's items. The overall QoL variable was then computed by summing the total scores of the six domains.

## Results

### Sociodemographic characteristics of respondents

In all, 124 PLWHA consented to participate, resulting in an 89% response rate. Table 1 shows the sociodemographic characteristics of the respondents. Most of the respondents (54.8%) were males. The mean age of the respondents was 35.38 years. Additionally, 58.0% of the respondents were never married, and 20.2% were divorced/widowed. Most of the respondents (62.1%) were Christians, and 14.5% were Muslims. Furthermore, more than half (72.6%) of the respondents were Akan, and 45.2% had attained secondary education. Almost all (80.6%) of the respondents were urban residents. In addition, more than half (58.9%) were employed. Of those employed, 20.5%, 21.9%, and 19.2% were traders, skilled workers, and artisan miners, respectively.

### Status disclosure and social support availability among PLWHA

Regarding status disclosure, 41.9% of the respondents disclosed their status to another person. Of these, a relative majority (46.2%) disclosed to their spouses, and 13.4% and 23.0% disclosed to children and nonfamily relations, respectively. Additionally, 76.9% of those who disclosed their HIV/AIDS status received a form of social support from such persons. Kinds of support received included support for medical expenses (47.5%), cash for upkeep (55.0%), shelter (52.5%), and emotional support (55.0%). The main sources of emotional support were spouses (60.0%), children (17.5%), parents (7.5%), and nonfamily relations (7.5%). Additionally, 16.9% of the respondents were in peer support groups (Table 2).

### Quality of life of people living with HIV/AIDS

This study also assessed the QoL of the respondents in Table 3. The mean QoL was highest within the physical domain (13.3±2.9), while the psychological domain indicated the lowest QoL among the respondents (11.03±3.26). The overall mean QoL score was 73.18, with 47.6% attaining at least this score.

Table 4 presents the results of a comparison of the means of QoL per background characteristics of the respondents. On average, respondents who were less than thirty years old had higher QoL in the physical, psychological, independence, social and environmental domains, and the difference was statistically significant. This same age group averaged a higher overall QoL, and the difference was statistically significant. Females recorded higher average QoL scores than males in physical, psychological, and independence domains as well as overall QoL, and the difference was statistically significant. Those who were currently married had higher average QoL scores in all domains, including overall QoL, and the difference was statistically significant. Regarding religion, Muslims had higher average QoL scores in all domains, including overall QoL, and the difference was statistically significant. Respondents with

**Table 1. Sociodemographic characteristics of respondents.**

| Sociodemographic variable | Frequency | Percentage (%) |
|---|---|---|
| **Sex** | | |
| Male | 68 | 54.8 |
| Female | 56 | 45.2 |
| **Age (in years)** | (Mean = 35.38±8.883) | |
| <30 | 32 | 25.8 |
| 30–39 | 52 | 41.9 |
| 40–49 | 30 | 24.2 |
| 50+ | 10 | 8.1 |
| **Marital status** | | |
| Never married | 72 | 58.0 |
| Married | 27 | 21.8 |
| Divorced/widowed | 25 | 20.2 |
| **Religion** | | |
| Christianity | 77 | 62.1 |
| Islam | 18 | 14.5 |
| None | 29 | 23.4 |
| **Ethnicity** | | |
| Akan | 90 | 72.6 |
| Ewe | 9 | 7.2 |
| Mole-Dagbani | 25 | 20.2 |
| **Educational level** | | |
| No formal education | 16 | 12.9 |
| Primary | 21 | 16.9 |
| Secondary | 56 | 45.2 |
| Tertiary | 31 | 25.0 |
| **Residence** | | |
| Rural | 24 | 19.4 |
| Urban | 100 | 80.6 |
| **Employment status** | | |
| Employed | 73 | 58.9 |
| Unemployed | 51 | 41.1 |
| **Occupation** | | |
| Trader | 15 | 20.5 |
| Farmer/breeder | 7 | 9.6 |
| Skilled work | 16 | 21.9 |
| Teacher | 12 | 16.5 |
| Unskilled labourer | 14 | 19.2 |
| Other | 9 | 12.3 |
| **Years since diagnosis** | (Mean = 12.35±5.97) | |
| <10 | 40 | 32.3 |
| 10–19 | 67 | 54.0 |
| 20+ | 17 | 13.7 |

tertiary education had a higher average QoL score in all domains, including overall QoL, and the difference was statistically significant. Even though respondents who disclosed their status had higher overall scores for all QoL domains except independence, the difference was not statistically significant.

**Table 2. Status disclosure and social support availability.**

| Variable | Frequency (N = 124) | Percentage (%) |
|---|---|---|
| **HIV/AIDS status disclosure** | | |
| Yes | 52 | 42 |
| No | 72 | 58 |
| **Person of first disclosure (n = 52)** | | |
| Spouse alone | 24 | 46.2 |
| Children alone | 7 | 13.4 |
| Spouse and children | 3 | 5.8 |
| Parents | 3 | 5.8 |
| Extended family | 3 | 5.8 |
| Nonfamily relations | 12 | 23.0 |
| **Availability of social support (n = 52)** | | |
| Yes | 40 | 76.9 |
| No | 12 | 23.1 |
| ***Kind of support received (n = 52)** | | |
| Funds for medical expenses | 19 | 47.5 |
| Cash for upkeep | 22 | 55.0 |
| Shelter | 21 | 52.5 |
| Support with food | 30 | 75.0 |
| Support for clothing | 18 | 45.0 |
| Emotional support | 22 | 55.0 |
| **Having a support group** | | |
| Yes | 21 | 16.9 |
| No | 103 | 83.1 |

*Multiple response

## Factors influencing quality of life among PLWHA

Table 5 presents the effects of status disclosure and other independent factors on the QoL of PLWHA. The regression model explained 51.5%, 54.0%, 55.6%, 41.2%, 48.7% and 44.4% of the variance in the physical, psychological, independence, social, environmental and spiritual domains, respectively. Overall, the regression model explained 51.8% of the variance in the overall quality of life score. As age increased, overall QoL decreased, and this observation was statistically significant. The QoL of males was significantly lower than that of females in all domains as well as overall QoL. Muslims had significantly higher QoL than Christians in all domains as well as overall QoL. Respondents who had tertiary education had significantly

**Table 3. Summary scores of WHOQOL-HIV domains.**

| QoL dimension | Mean score | Minimum score | Maximum score | 95% confidence interval |
|---|---|---|---|---|
| Physical | 13.63 | 4.00 | 19.00 | 12.98–14.28 |
| Psychological | 11.03 | 4.80 | 17.60 | 10.45–11.61 |
| Independence | 11.94 | 7.00 | 16.00 | 11.34–12.44 |
| Social | 12.83 | 7.00 | 19.00 | 12.29–13.37 |
| Environmental | 11.38 | 8.00 | 16.00 | 11.09–11.72 |
| Spiritual | 12.38 | 9.00 | 19.00 | 11.95–12.81 |
| Overall QoL | 73.18 | 43.30 | 101.80 | 70.51–75.84 |

**Table 4. Comparison of mean QoL scores for background characteristics.**

| Variable | QoL domains | | | | | | |
|---|---|---|---|---|---|---|---|
| | Physical | Psychological | Independence | Social | Environmental | Spiritual | Overall QoL |
| | Mean(S.D.) | Mean(S.D.) | Mean(S.D.) | Mean(S.D.) | Mean(S.D.) | Mean(S.D.) | Mean(S.D.) |
| **Age group** | | | | | | | |
| <30 | 16.03(2.33) | 12.70(2.32) | 14.06(1.54) | 14.44(2.65) | 11.94(1.42) | 12.22(1.58) | 81.39(8.87) |
| 30–39 | 13.12(3.22) | 10.65(3.03) | 11.81(2.60) | 12.08(2.56) | 11.03(1.95) | 12.33(2.53) | 71.00(13.89) |
| 40–49 | 13.20(4.50) | 10.88(4.06) | 11.20(2.87) | 13.50(3.38) | 11.85(2.31) | 12.90(3.20) | 73.53(18.90) |
| 50+ | 9.90(1.85) | 8.08(1.16) | 8.00(0.82) | 9.60(1.71) | 9.95(1.09) | 11.60(0.52) | 57.13(2.62) |
| *P value* | <0.001 | <0.001 | <0.001 | <0.001 | <0.01 | 0.462 | <0.001 |
| **Sex** | | | | | | | |
| Female | 17.71(2.96) | 11.79(3.01) | 13.00(2.39) | 13.41(2.55) | 11.57(1.61) | 12.27(2.27) | 76.75(12.25) |
| Male | 12.74(3.96) | 10.40(3.35) | 11.06(2.86) | 12.35(3.34) | 11.21(2.18) | 12.47(2.54) | 70.23(16.44) |
| *P value* | <0.01 | <0.05 | <0.001 | 0.054 | 0.309 | 0.644 | <0.05 |
| **Marital status** | | | | | | | |
| Never married | 14.01(2.74) | 11.03(2.50) | 12.61(2.38) | 12.81(2.74) | 11.06(1.38) | 11.83(1.95) | 73.36(11.23) |
| Currently married | 15.22(4.82) | 13.16(4.38) | 13.00(2.50) | 14.22(3.56) | 12.89(2.20) | 14.56(2.81) | 83.04(18.96) |
| Divorced/widowed | 10.80(3.16) | 8.70(2.13) | 8.84(2.15) | 11.40(2.69) | 10.64(2.25) | 11.60(1.80) | 61.98(12.26) |
| *P value* | <0.001 | <0.001 | <0.001 | <0.01 | <0.001 | <0.001 | <0.001 |
| **Religion** | | | | | | | |
| Christianity | 14.03(3.70) | 11.26(3.11) | 12.31(2.76) | 12.82(3.00) | 11.39(1.65) | 12.17(2.26) | 73.98(14.34) |
| Islam | 15.17(3.28) | 13.07(3.82) | 13.00(2.06) | 15.17(2.18) | 12.67(2.05) | 14.83(2.47) | 83.90(14.63) |
| Other | 11.62(3.05) | 9.13(2.26) | 10.28(2.78) | 11.41(2.80) | 10.53(2.20) | 11.41(1.74) | 64.39(12.02) |
| *P value* | <0.01 | <0.001 | <0.01 | <0.001 | <0.01 | <0.001 | <0.001 |
| **Ethnicity** | | | | | | | |
| Akan | 13.63(3.64) | 10.93(3.09) | 11.79(2.84) | 12.53(2.99) | 11.48(1.75) | 12.30(2.26) | 72.67(14.40) |
| Ewe | 12.67(4.27) | 10.40(2.50) | 11.67(3.61) | 12.67(3.28) | 10.33(1.75) | 10.67(0.50) | 68.40(14.34) |
| Mole-Dagbani | 13.96(3.65) | 11.58(4.08) | 12.56(2.47) | 13.96(3.01) | 11.36(2.56) | 13.28(2.99) | 76.70(17.13) |
| *P value* | 0.666 | 0.571 | 0.465 | 0.115 | 0.241 | <0.05 | 0.304 |
| **Educational level** | | | | | | | |
| None | 11.88(3.77) | 10.75(3.10) | 11.19(3.49) | 12.06(3.75) | 9.88(1.96) | 12.31(2.44) | 68.06(16.74) |
| Primary | 11.86(4.00) | 9.37(2.62) | 11.43(2.62) | 11.71(2.61) | 11.21(2.05) | 12.00(2.05) | 67.59(14.37) |
| Secondary | 13.52(3.54) | 10.67(3.32) | 11.50(3.03) | 12.72(2.74) | 11.04(1.47) | 11.63(1.97) | 71.10(13.57) |
| Tertiary | 15.94(2.26) | 12.93(2.84) | 13.45(1.39) | 14.13(3.12) | 12.87(1.75) | 14.03(2.65) | 83.35(12.50) |
| *P value* | <0.001 | <0.01 | <0.01 | <0.05 | <0.001 | <0.001 | <0.001 |
| **Residence** | | | | | | | |
| Rural | 12.88(3.37) | 9.60(1.42) | 11.50(2.93) | 11.25(2.64) | 11.06(1.43) | 11.63(1.53) | 67.91(10.79) |
| Urban | 13.81(3.73) | 11.37(3.48) | 12.04(2.80) | 13.21(3.02) | 11.45(2.05) | 12.56(2.56) | 74.44(15.63) |
| *P value* | 0.264 | <0.05 | 0.402 | <0.01 | 0.383 | 0.089 | 0.055 |
| **Employment** | | | | | | | |
| Unemployed | 13.77(3.84) | 10.45(3.29) | 12(59(2.48) | 12.75(3.08) | 10.68(1.60) | 11.82(1.90) | 72.05(14.16) |
| Trader/business person | 12.80(1.90) | 11.36(2.31) | 11.60(2.50) | 13.00(1.73) | 11.50(0.93) | 12.20(1.21) | 72.46(7.28) |
| Farmer/breeder | 10.29(2.14) | 8.80(0.00) | 7.57(0.53) | 8.57(0.53) | 10.57(0.53) | 11.43(0.53) | 57.23(3.21) |
| Skilled worker | 11.00(3.37) | 9.95(2.85) | 9.88(2.42) | 13.13(2.55) | 10.91(1.90) | 11.50(2.31) | 66.36(13.92) |
| Teacher | 16.75(2.38) | 13.60(3.34) | 14.00(1.95) | 13.75(3.72) | 13.00(1.92) | 14.25(2.99) | 85.35(14.24) |
| Unskilled labourer | 14.93(4.07) | 12.46(4.30) | 11.79(3.42) | 13.86(3.32) | 11.96(2.70) | 13.86(3.74) | 78.85(20.56) |
| Other | 15.33(1.80) | 11.73(1.74) | 13.33(1.00) | 13.00(3.00) | 13.50(1.73) | 13.33(2.18) | 80.23(10.87) |
| *P value* | <0.001 | <0.01 | <0.001 | <0.01 | <0.001 | <0.01 | <0.001 |
| **Years since diagnosis** | | | | | | | |

*(Continued)*

**Table 4.** (Continued)

| Variable | QoL domains | | | | | | | Overall QoL |
|---|---|---|---|---|---|---|---|
| | Physical | Psychological | Independence | Social | Environmental | Spiritual | Overall QoL |
| | Mean(S.D.) | Mean(S.D.) | Mean(S.D.) | Mean(S.D.) | Mean(S.D.) | Mean(S.D.) | Mean(S.D.) |
| **Age group** | | | | | | | |
| <10 | 14.25(3.23) | 11.12(3.22) | 12.35(2.43) | 12.25(2.62) | 11.33(1.69) | 12.58(2.74) | 73.87(13.63) |
| 10–19 | 14.25(3.40) | 11.79(3.07) | 12.40(2.79) | 13.81(2.91) | 11.78(1.96) | 12.51(2.43) | 76.53(14.45) |
| 20+ | 9.71(3.37) | 7.81(2.05) | 9.12(2.23) | 10.35(2.87) | 9.91(1.80) | 11.41(0.94) | 58.31(11.52) |
| *P value* | <0.001 | <0.001 | <0.001 | <0.001 | <0.01 | 0.205 | <0.001 |
| **Peer support group member** | | | | | | | |
| No | 13.60(3.56) | 10.72(3.05) | 11.85(2.81) | 12.64(2.83) | 11.17(1.71) | 12.21(2.39) | 72.18(14.01) |
| Yes | 13.76(4.25) | 12.53(3.90) | 12.38(2.92) | 13.76(3.86) | 12.41(2.65) | 13.19(2.44) | 78.03(18.80) |
| *P value* | 0.856 | <0.05 | 0.430 | 0.124 | <0.01 | 0.091 | 0.104 |
| **HIV/AIDS status disclosure** | | | | | | | |
| No | 13.35(3.37) | 10.90(2.92) | 12.24(2.61) | 12.85(3.03) | 11.08(1.82) | 12.04(2.09) | 72.46(13.42) |
| Yes | 14.02(4.05) | 11.20(3.71) | 11.52(3.07) | 12.81(3.09) | 11.78(2.06) | 12.85(2.75) | 74.17(17.03) |
| *P value* | 0.316 | 0.615 | 0.164 | 0.943 | 0.049* | 0.067 | 0.532 |

S.D = Standard Deviation

better QoL than those without any education. On the other hand, respondents who had tertiary education recorded lower QoL in the psychological domain than those who had no formal education, and this difference was statistically significant. Respondents who lived in urban areas had better QoL in all domains and overall QoL than those who lived in rural areas, and the difference was statistically significant. Although respondents who disclosed their status recorded better QoL overall than those who did not disclose their status, the difference was not statistically significant.

## Discussions

We examined HIV/AIDS status disclosure and social support availability and their influence on quality of life among PLWHA in the Obuasi East District of the Ashanti Region, Ghana. We found that less than half of the HIV/AIDS clients had disclosed their status to someone such as a spouse, children, and nonfamily relations. Additionally, most of the clients had moderate quality of life across most of the six domains. We did not find a significant link between HIV/AIDS status disclosure and quality of life in any of the domains.

The finding that most HIV/AIDS clients did not disclose their status is congruent with many studies across the globe. Our status disclosure of 41.9% is consistent with the argument by Ssali et al. that HIV/AIDS status disclosure ranges from 24–91% [17] and is lower than the 49% reported by the WHO in the developing world [42]. Our finding could be attributed to the high levels of stigma and discrimination [43], fear of severed relationships, abuse [44], and feelings of shame in certain settings [45]. The finding suggests a need for interventions aimed at reducing stigma and discrimination to encourage more individuals to disclose their status, thus potentially improving overall QoL. Strategies for reducing stigma and discrimination among PLWHA may involve community education campaigns, workplace sensitivity training, and legal protections against discrimination based on HIV status.

We found in our study that HIV-positive patients had moderate QoL scores in all domains. This supports the argument that being on ART leads to better QoL, regardless of the duration of the ART [46]. It is therefore expected that most of the clients in this study would have better

**Table 5. Effect of status disclosure on factors influencing quality of life among PLHIV.**

| Variable | QoL domains | | | | | | Overall QoL |
|---|---|---|---|---|---|---|---|
| | Physical | Psychological | Independence | Social | Environmental | Spiritual | |
| | β(95%CI) | β(95%CI) | β(95%CI) | β(95%CI) | β(95%CI) | β(95%CI) | β(95%CI) |
| Constant | 21.63(17.2, 26.1)*** | 20.75(16.9, 24.6)*** | 18.41(15.1, 21.7)*** | 19.99(15.9, 24.0)*** | 12.41(10.0, 14.8)*** | 15.61(12.5, 18.7)*** | 108.81(90.7, 126.9)*** |
| Age (in years) | -0.23(-0.3, -0.1)*** | -0.24(-0.3, -0.2)*** | -0.18(-0.3, -0.1)*** | -0.25(-0.3, -0.1)*** | -0.07(-0.2, -0.1)* | -0.07(-0.1, 0.0) | -1.03(-1.5, -0.6)*** |
| **Sex** | | | | | | | |
| Female | 1 | 1 | 1 | 1 | 1 | 1 | 1 |
| Male | -2.81(-4.4, -1.2)** | -2.82(-4.2, -1.4)*** | -1.97(-3.1, -0.8)** | -2.23(-3.7, -0.8)** | -1.83(-2.7, -1.0)*** | -1.47(-2.6, -0.4)* | -13.14(-19.6, -6.7)*** |
| **Marital status** | | | | | | | |
| Never married | 1 | 1 | 1 | 1 | 1 | 1 | 1 |
| Currently married | 0.28(-1.6, 2.1) | 1.81(0.2, 3.4)* | -0.01(-1.4, 1.4) | 1.43(-0.3, 3.1) | 0.52(-0.5, 1.5) | 2.29(1.0, 3.6)** | 6.32(-1.3, 13.9) |
| Divorced/widowed | 1.17(-1.4, 3.7) | 2.41(0.2, 4.6)* | -0.88(-2.8, 1.0) | 3.20(0.8, 5.6)** | 0.80(-0.6, 2.2) | 2.47(0.7, 4.3)** | 9.17(-1.3, 19.7) |
| **Religion** | | | | | | | |
| Christianity | 1 | 1 | 1 | 1 | 1 | 1 | 1 |
| Islam | 5.08(2.8, 7.3)*** | 4.73(2.8, 6.7)*** | 3.69(2.0, 5.4)*** | 5.12(3.1, 7.2)*** | 3.10(1.9, 4.3)*** | 3.86(2.3, 5.4)*** | 25.58(16.4, 34.8)*** |
| Other | 1.54(-0.6, 3.6) | -0.98(-2.8, 0.8) | 1.87(0.3, 3.4)* | 0.42(-1.5, 2.3) | 0.57(-0.6, 1.7) | -0.26(-1.7, 1.2) | 3.16(-5.4, 11.7) |
| **Ethnicity** | | | | | | | |
| Akan | 1 | 1 | 1 | 1 | 1 | 1 | 1 |
| Ewe | -2.19(-4.4, 0.0) | -2.18(-4.1, -0.3)* | -1.08(-2.7, 0.5) | -2.29(-4.3, -0.3)* | -0.82(-2.0, 0.4) | -1.84(-3.4, -0.3)* | -10.39(-19.3, -1.4)* |
| Mole-Dagbani | -2.07(-4.0, -0.1)* | -1.96(-3.6, -0.3)* | -0.88(-2.3, 0.5) | -1.47(-3.2, 0.3) | -1.25(-2.3, -0.2)* | -1.63(-3.0, -0.3)* | -9.26(-17.1, -1.4)* |
| **Educational level** | | | | | | | |
| None | 1 | 1 | 1 | 1 | 1 | 1 | 1 |
| Primary | -2.28(-4.7, 0.1) | -4.17(-6.2, -2.1)*** | -1.00(-2.8, 0.8) | -1.65(-3.8, 0.5) | 1.01(-0.3, 2.3) | -1.72(-3.4, -0.1)* | -9.80(-19.3, -1.4)* |
| Secondary | -1.49(-3.5, 0.5) | -4.17(-5.9, -2.4)*** | -1.55(-3.0, -0.1)* | -2.12(-4.0, -0.3)* | 0.20(-0.9, 1.3) | -2.60(-4.0, -1.2)*** | -11.73(-20.0, -3.5)** |
| Tertiary | -0.12(-2.6, 2.3) | -3.14(-5.3, -1.0)** | -0.52(-2.3, 1.3) | -0.93(-3.2, 1.3) | 1.35(0.0, 2.7)* | -1.26(-3.0, 0.5) | -4.63(-14.6, 5.4) |
| **Residence** | | | | | | | |
| Rural | 1 | 1 | 1 | 1 | 1 | 1 | 1 |
| Urban | 2.11(0.8, 3.4)** | 2.44(1.3, 3.6)*** | 1.07(0.1, 2.0)* | 2.31(1.1, 3.5)*** | 0.77(0.1, 1.5)*. | 1.24(0.3, 2.2)** | 9.93(4.6, 15.2)*** |
| **Employment** | | | | | | | |
| Unemployed | 1 | 1 | | 1 | 1 | 1 | 1 |
| Trader/business person | -1.38(-3.4, 0.6) | -0.78(-2.5, 0.9) | -0.98(-2.4, 0.5) | -0.98(-2.8, 0.8) | -0.60(-1.7, 0.5) | -1.73(-3.1, -0.3)* | -6.46(-14.5, 1.6) |
| Farmer/breeder | -0.24(-3.3, 2.8) | 3.01(0.4, 5.7)* | -1.34(-3.6, 0.9) | -1.75(-4.6, 1.0) | 1.83(0.2, 3.5)* | -0.09(-2.3, 2.1) | 1.42(-11.1, 13.9) |
| Skilled worker | -1.72(-3.6, 0.2) | 0.93(-0.7, 2.6) | -1.63(-3.0, -0.2)* | 1.12(-0.6, 2.8) | 0.97(-0.1, 2.0) | -0.57(-1.9, 0.8) | -0.90(-8.6, 6.8) |
| Teacher | 2.29(0.3, 4.3)* | 2.33(0.6, 4.1)* | 1.56(0.1, 3.0)* | 0.56(-1.3, 2.4) | 1.71(0.6, 2.8)** | 1.15(-0.2, 2.6) | 9.58(1.4, 17.8)* |
| Unskilled labourer | 2.40(0.5, 4.2)* | 3.01(1.4, 4.6)*** | 0.26(-1.1, 1.6) | 1.41(-0.3, 3.1) | 1.55(0.5, 2.6)** | 1.32(0.0, 2.6)* | 9.94(2.4, 17.5)* |
| Other | 3.47(1.1, 5.9)** | 3.41(1.3, 5.5)** | 2.51(0.7, 4.3)** | 0.92(-1.3, 3.1) | 2.81(1.5, 4.1)*** | 0.01(-1.7, 1.7) | 13.12(3.3, 22.9)** |
| **Years since diagnosis** | -0.05(-0.2, 0.1) | 0.01(-0.1, 0.1) | 0.04(-0.1, 0.1) | 0.07(-0.1, 0.2) | 0.00(-0.1, 0.1) | -0.03(-0.1, 0.1) | 0.04(-0.4, 0.5) |
| **Peer support group member** | | | | | | | |
| No | 1 | 1 | 1 | 1 | 1 | 1 | 1 |
| Yes | -0.47(-1.9, 1.0) | 0.74(-0.5, 2.0) | 0.24(-0.8, 1.3) | -0.09(-1.4, 1.2) | 0.72(-0.1, 1.5) | 0.50(-0.5, 1.5) | 1.64(-4.2, 7.5) |
| **HIV/AIDS status disclosure** | | | | | | | |
| No | 1 | 1 | 1 | 1 | 1 | 1 | 1 |
| Yes | 1.17(-0.1, 2.5) | -0.25(-1.4, 0.9) | -0.02(-1.0, 0.9) | -0.10(-1.3, 1.1) | -0.01(-0.7, 0.7) | -0.12(-1.0, 0.8) | 0.67(-4.6, 6.0) |
| **Adjusted R square** | 0.515 | 0.540 | 0.556 | 0.412 | 0.487 | 0.444 | 0.518 |

*p<0.05

**p<0.01

***p<0.001; 1 = Reference category

QoL similar to those in other low-income countries [47–49]. This is because PLWHA attend routine ART clinics for their follow-up and receive counselling, care and treatment, which may have positively impacted all the domains of health-related QoL. Additional support or interventions may thus be needed to enhance QoL beyond medical treatment alone.

Our finding that disclosure of HIV/AIDS status was not a statistically significant predictor of quality of life is supported by a study by Agbeko et al. [50], who found that disclosure status of HIV/AIDS among PLWHA in the Ho municipality of Ghana was not statistically significant, although disclosure improved the overall QoL of PLWHA. Similarly, the findings of Chandra and colleagues that HIV serostatus disclosure was not significantly associated with QoL gives credence to our result [51]. Other studies, by contrast, established a statistically significant relationship between disclosure of HIV/AIDS status and quality of life [46, 52, 53]. Even though our finding was not statistically significant, the fact that disclosure improves the QoL of HIV/AIDS could be explained by the role of supportive relationships minimizing stigmatization and discrimination toward PLWHA, which consequently boosts the QoL of PLWHA. Supportive social networks have been shown to play a crucial role in buffering the negative impact of HIV/AIDS-related stigma and discrimination, thereby promoting psychological well-being and overall QoL among PLWHA. Additionally, policies and interventions aimed at reducing stigma and discrimination within communities could further amplify the positive effects of disclosure on QoL. Thus, while our study did not find a direct statistical link, the broader context suggests that disclosure may indeed contribute to improved QoL through its influence on social support and stigma reduction. Social support networks could be enhanced through establishing peer support groups, providing counselling services, and promoting community engagement through awareness-raising events and activities.

Furthermore, we discovered that age negatively predicted the overall quality of life of PLWHA to a statistically significant degree. In other words, as PLWHA grow older, their QoL becomes poorer. This finding is analogous to a Kazakhstan study that established that increasing age decreases the QoL of PLWHA [54]. In the same vein, Chinese and Pakistani studies found that age was negatively associated with QoL [55, 56]. Some authors have made contradictory findings, however. For instance, studies in Burkina Faso showed that older PLWHA were more likely to have higher QoL than younger PLWHA [49]. There seems to be a logical explanation for our finding on the relationship between age and QoL. As people grow older, they are more exposed to diseases that weaken their immune system, thus compounding their healthcare problems and undercutting their QoL. Likewise, mental health challenges are commonly associated with old age.

We also established that female PLWHA had significantly higher QoL in all domains than their male counterparts. This finding is partially supported by a study that showed that females had significantly higher QoL in the domain of physical well-being than males [57]. Other studies, however, suggest that being a male PLWHA is a positive predictor of QoL [49, 56, 58, 59]. Meanwhile, some researchers did not find a statistically significant relationship between QoL and the sex of PLWHA [13, 50]. There is no consensus in the literature on the influence of sex on the QoL of PLWHA. Both groups of researchers who found opposing relationships between sex and QoL of PLWHA rely on one explanation for their observations: gender roles. Gender roles resign females living with HIV/AIDS to discriminatory behaviour, which could hinder their QoL. Additionally, gender roles limit access to the financial resources of these females, which impedes their ability to fend for themselves and improve their QoL. The same gender roles hurt men living with HIV/AIDS, too, by putting financial stress on men—in their role as breadwinners of the family, which invariably affects their QoL.

Our study also pointed to the influence of geographic location on the QoL of PLWHA. Urban residency is a positive predictor of HIV/AIDS. Although studies are largely silent on the impact of geographical location on the QoL of PLWHA, the study by Shriharsha and Rentala [58] found that PLWHA in urban areas had significantly higher QoL, supporting our findings. Urban locations, especially in developing settings such as Ghana, provide an advantage regarding physical access to ART services, which empowers PLWHA and puts them in a better position to cope with the disease than their colleagues in rural areas. Improving access to healthcare services for PLWHA in different demographic groups may require initiatives such as mobile clinics for rural areas, transportation subsidies, and culturally sensitive healthcare provider training programs.

In addition to the geographical area, our findings revealed the influence of religion on the QoL of PLWHA. Belonging to the Islamic religion was a positive predictor of quality of life. The influence of religion on the QoL of PLWHA is an established concept [60, 61]. Religious bodies present an opportunity for social support for PLWHA. However, the particular reason for the advantage that the Islamic religion gives to PLWHA is unclear. Perhaps Islamic practices are more collegial and community-centred than Christian practices.

Moreover, the intersectionality of sociodemographic factors, including age, gender, geographic location, and religion, presents a complex interplay that influences the quality of life of PLWHA. In the Ghanaian context, cultural norms, access to resources, and community support systems may interact with these sociodemographic factors to shape the experiences and well-being of PLWHA. For example, older PLWHA in rural areas may face compounded challenges due to limited access to healthcare services and social support networks, leading to a significant impact on their QoL compared to younger counterparts in urban settings. Gender dynamics further complicate this scenario, as cultural norms and expectations may influence access to employment opportunities and financial resources differently for men and women living with HIV/AIDS. Additionally, geographic location plays a pivotal role; while urban areas typically offer better access to healthcare facilities and support services, factors such as stigma and discrimination may still persist, particularly affecting QoL among PLWHA in densely populated urban centres. Moreover, religious affiliation can influence social support structures and coping mechanisms, with religious communities often providing essential support networks for PLWHA. For instance, mosques and churches may serve as hubs for community gatherings and support groups, facilitating emotional support and enhancing QoL for PLWHA. Understanding these complex interactions is crucial for tailoring targeted interventions and policies to address the diverse needs of PLWHA across different sociodemographic groups. Exploring potential barriers or challenges to implementing these interventions could involve assessing cultural beliefs, addressing healthcare infrastructure limitations, and overcoming financial constraints for both individuals and healthcare systems. Collaborating with local community leaders, leveraging technology for remote healthcare delivery, and advocating for policy changes to support comprehensive HIV/AIDS care can help overcome these barriers.

## Strengths and limitations

This study derives its strength from the use of a standard survey instrument in assessing the quality of life of PLWHA. The adoption of higher-level statistical analytical approaches (multiple linear regression) in this study enhances the accuracy of the findings and allows for a deeper understanding of the relationships between variables. Despite these strengths, several limitations must be acknowledged to contextualize the interpretation and generalizability of the findings. The relatively small sample size restricts the extent to which the results can be

extrapolated to broader populations of PLWHA. Again, the quantitative nature of the study limits the robustness of the findings, as some important sociocultural factors that may also influence the quality of life of PLWHA can only be assessed qualitatively. Also, focusing on only one health facility in a municipality in Ghana raises concern about the generalizability of the findings to other health facilities within the municipality and the country as a whole. Variations in healthcare infrastructure, resources, and patient demographics across different facilities may impact the applicability of the findings to broader contexts. Finally, the reliance on self-reported data introduces the possibility of social desirability bias, where participants may provide responses that align with societal expectations rather than their true experiences. While efforts were made to mitigate this bias through confidentiality and anonymity, its potential influence on the study findings should be acknowledged.

## Conclusions

The finding that less than half of the PLWHA disclosed their HIV/AIDS status implies that most of their sexual and nonsexual relations are at risk of infection. Additionally, this suggests that persistent stigma and discrimination may contribute to the reluctance of most PLWHA to disclose their status. This may affect the fight to reduce the infection and improve the quality of life for PLWHA.

Moreover, the finding that most PLWHA had moderate quality of life implies that ART services likely have a positive impact on the clients' well-being. A good quality of life suggests improvement in the overall health of HIV/AIDS patients, potentially reducing morbidity and mortality associated with the disease. Ghana's chances of attaining goal 3.3 of the Sustainable Development Goals, which seeks to end the HIV/AIDS epidemic and reduce mortality associated with the condition, could be increased by stakeholders through initiatives that target and limit the influence of negative predictors of the quality of life of PLWHA.

Based on the implications of these findings, we recommend that the Ghana AIDS Commission and the Ghana Health Service intensify efforts to reduce discrimination and stigmatization associated with HIV/AIDS. Furthermore, stakeholders in AIDS prevention in Ghana should employ innovative methods such as peer support to encourage status disclosure and provide education on the provision of social and emotional support for PLWHA.

## Supporting information

**S1 Data. Study data.**
(SAV)

## Author Contributions

**Conceptualization:** Farrukh Ishaque Saah, Abdul-Aziz Seidu, Hubert Amu, Luchuo Engelbert Bain.

**Data curation:** Farrukh Ishaque Saah.

**Formal analysis:** Farrukh Ishaque Saah.

**Funding acquisition:** Farrukh Ishaque Saah, Abdul-Aziz Seidu.

**Investigation:** Farrukh Ishaque Saah, Kizito Aidam.

**Methodology:** Farrukh Ishaque Saah, Abdul-Aziz Seidu, Hubert Amu, Luchuo Engelbert Bain.

**Project administration:** Farrukh Ishaque Saah.

**Resources:** Farrukh Ishaque Saah, Hubert Amu.

**Software:** Hubert Amu.

**Supervision:** Farrukh Ishaque Saah, Abdul-Aziz Seidu, Hubert Amu, Luchuo Engelbert Bain.

**Validation:** Farrukh Ishaque Saah, Kizito Aidam, Abdul-Aziz Seidu, Hubert Amu, Luchuo Engelbert Bain.

**Visualization:** Farrukh Ishaque Saah.

**Writing – original draft:** Farrukh Ishaque Saah, Kizito Aidam.

**Writing – review & editing:** Farrukh Ishaque Saah, Kizito Aidam, Abdul-Aziz Seidu, Hubert Amu, Luchuo Engelbert Bain.

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
