## [Decision Letter · Decision Letter 0]

14 Jan 2024

PMEN-D-23-00040

Effect of Status Disclosure on Quality of Life among People Living with HIV/AIDS in Ghana: A Health Facility-based Cross-sectional Study

PLOS Mental Health

Dear Dr. Saah,

Thank you for submitting your manuscript to PLOS Mental Health. After careful consideration, we feel that it has merit but does not fully meet PLOS Mental Health’s publication criteria as it currently stands. Therefore, we invite you to submit a revised version of the manuscript that addresses the points raised during the review process.

We look forward to receiving your revised manuscript.

Kind regards,

Isabella Giulia Franzoi, Ph.D.

Academic Editor

PLOS Mental Health

Journal Requirements:

1. In the online submission form, you indicated that "Data is made available upon reasonable request from the corresponding author". All PLOS journals now require all data underlying the findings described in their manuscript to be freely available to other researchers, either 1. In a public repository, 2. Within the manuscript itself, or 3. Uploaded as supplementary information.

Additional Editor Comments (if provided):

The study investigates a crucial public health topic, that is still mainly understudied. Namely, the HIV/AIDS status disclosure and its impact on quality of life. Since there is a limited number of studies addressing this issue in Ghana, this study appears to be important, despite its limited numbers. however, as indicated by the reviewers, it has some important critical issues that need to be resolved in order for it to become publishable

Reviewers' comments:

Reviewer's Responses to Questions

**Comments to the Author**

1. Does this manuscript meet PLOS Mental Health’s publication criteria? Is the manuscript technically sound, and do the data support the conclusions? The manuscript must describe methodologically and ethically rigorous research with conclusions that are appropriately drawn based on the data presented.

Reviewer #1: Yes

Reviewer #2: No

Reviewer #3: Partly

2. Has the statistical analysis been performed appropriately and rigorously?

Reviewer #1: No

Reviewer #2: Yes

Reviewer #3: Yes

3. Have the authors made all data underlying the findings in their manuscript fully available (please refer to the Data Availability Statement at the start of the manuscript PDF file)?

Reviewer #1: Yes

Reviewer #2: No

Reviewer #3: Yes

4. Is the manuscript presented in an intelligible fashion and written in standard English?

Reviewer #1: Yes

Reviewer #2: Yes

Reviewer #3: Yes

5. Review Comments to the Author

Reviewer #1: General comments

The study investigated an important public health issue, from the perspective of resource-limited setting. However, I have itemized my comments, concerns, and suggestions below.

Title

The title is clear and concise.

Abstract

The abstract is good summary of the whole manuscript, including the results presented.

Introduction

The introduction is well written. The research problem and study rationale were clearly stated by the authors.

Methods and materials

Study site and population

The information on the study site appears scanty. The authors should please kindly provide additional information on the study setting as related to HIV/AIDS treatment in particular and healthcare in general.

Sampling

The sample size calculation does not seem right, especially the population size used for the calculation. This made the sample size required for the study appear rather too small. Using the total number of PLWHA as contained on the register of the HIV clinics selected for the study would have been more appropriate. The small sample size in this study will limit the application of its findings beyond the study group.

Data analysis

Linear regression does not seem appropriate for the dataset. Normality is not the only requirement for running linear regression test. The dependent and independent variables must have a linear relationship. Both dependent and independent variables must be measured at the continuous level. I do not think the variables in this study, especially the independent variables fulfilled these requirements. In the study, the independent variables are categorized and are nominal data. The authors should consider conducting chi-square test followed by binary logistic regression. In doing so, the authors should first classify the health-related quality of life scores into two groups (good & poor).

Discussion

The findings are well discussed in view existing literature. However, the authors should give attention to the concerns raised above.

Conclusion

The conclusion is based on the results presented.

Reviewer #2: This study addresses a pivotal subject matter, focusing on the impact of status disclosure on the quality of life (QoL) of individuals living with HIV/AIDS (PLWHA). However, there is a suggestion that employing a qualitative or case-control study design would have been more advantageous than the chosen cross-sectional approach.

In assessing the effects of status disclosure on QoL, how did the authors precisely determine the time elapsed since disclosure (in days, weeks, or months) and categorize participants as "hospitalized" or "outpatients"? Additionally, what methodology was employed to conclude that the specific aspect of "status disclosure" significantly affected participants' quality of life based on events occurring in the past month, utilizing the WHOQOL Bref instrument?

The exclusion of seriously ill PLWHA from the study is mentioned. Could the authors provide clarification on the criteria used to classify the status of patients, leading to their exclusion?

The study suggests that qualitative methods are superior for assessing quality of life, yet a cross-sectional survey design was chosen. Can the authors elaborate on the rationale behind this choice, especially considering the acknowledged importance of the subject matter?

Reviewer #3: Reviewer comments

The study addresses the crucial topic of HIV/AIDS status disclosure and its potential impact on the quality of life (QoL) among People Living with HIV/AIDS (PLWHA) in Ghana. Given the limited number of studies specifically addressing QoL among PLWHA in Ghana, this study is deemed important. Even with a small and non-representative sample, the study holds value.

POINT -1 The introduction effectively emphasizes the importance of the well-being of individuals with HIV. Nevertheless, the text emphasizes the benefits without sufficiently addressing the complex socio-cultural factors contributing to the quality of life for individuals. I suggest including information about stigma and discrimination, particularly in the context of sub-Saharan Africa and Ghana. In the ethics section, the authors must include the ethics committee approval number.

POINT 2 - The method used is adequate, following the STROBE guide. The statistical method used is suitable for epidemiological studies.

POINT 3- The study provides valuable insights into the low rate of HIV/AIDS status disclosure among PLWHA in the Obuasi East District, aligning with global trends. While the study identifies several predictors of quality of life (QoL) among PLWHA, it lacks a proposed cause for the non-significant link between disclosure and QoL. I suggest reinforcing the importance of factors influencing QoL, such as supportive relationships and reduced stigma. The discussion presents intriguing findings regarding the impact of age, gender, geographic location, and religion on the QoL of PLWHA. However, the study could benefit from exploring how age, gender, location, and religion may interact to shape the experiences and well-being of PLWHA. Other countries yield results on these sociodemographic characteristics.

POINT 4- In future directions, there is room for a more nuanced discussion regarding the implications of the study's findings for healthcare policies and interventions in Ghana. Further exploration of the factors contributing to the observed disparities in QoL, especially socio-demographic predictors, could offer valuable insights for targeted interventions aimed at improving the well-being of PLWHA in different demographic groups. Additionally, it would be beneficial to highlight the limitations of the study more explicitly, such as the potential biases associated with a cross-sectional design in the specific setting, to enhance the interpretation and generalizability of the findings. It is suggested to reiterate information about the scenario in the realized study and its representation in Ghana.

6. PLOS authors have the option to publish the peer review history of their article (what does this mean?). If published, this will include your full peer review and any attached files.

**Do you want your identity to be public for this peer review?** For information about this choice, including consent withdrawal, please see our Privacy Policy.

Reviewer #1: **Yes: **Chibueze Anosike

Reviewer #2: **Yes: **Ahmad Neyazi

Reviewer #3: No

---

## [Decision Letter · Decision Letter 1]

17 Apr 2024

PMEN-D-23-00040R1

Effect of Status Disclosure on Quality of Life among People Living with HIV/AIDS in Ghana: A Health Facility-based Cross-sectional Study

PLOS Mental Health

Dear Dr. Saah,

Thank you for submitting your manuscript to PLOS Mental Health. After careful consideration, we feel that it has merit but does not fully meet PLOS Mental Health’s publication criteria as it currently stands. Therefore, we invite you to submit a revised version of the manuscript that addresses the points raised during the review process.

We look forward to receiving your revised manuscript.

Kind regards,

Isabella Giulia Franzoi, Ph.D.

Academic Editor

PLOS Mental Health

Journal Requirements:

Additional Editor Comments (if provided):

Reviewers' comments:

Reviewer's Responses to Questions

**Comments to the Author**

1. If the authors have adequately addressed your comments raised in a previous round of review and you feel that this manuscript is now acceptable for publication, you may indicate that here to bypass the “Comments to the Author” section, enter your conflict of interest statement in the “Confidential to Editor” section, and submit your "Accept" recommendation.

Reviewer #4: All comments have been addressed

Reviewer #5: All comments have been addressed

Reviewer #6: (No Response)

2. Does this manuscript meet PLOS Mental Health’s publication criteria? Is the manuscript technically sound, and do the data support the conclusions? The manuscript must describe methodologically and ethically rigorous research with conclusions that are appropriately drawn based on the data presented.

Reviewer #4: Partly

Reviewer #5: Yes

Reviewer #6: Partly

3. Has the statistical analysis been performed appropriately and rigorously?

Reviewer #4: N/A

Reviewer #5: Yes

Reviewer #6: Yes

4. Have the authors made all data underlying the findings in their manuscript fully available (please refer to the Data Availability Statement at the start of the manuscript PDF file)?

Reviewer #4: Yes

Reviewer #5: Yes

Reviewer #6: No

5. Is the manuscript presented in an intelligible fashion and written in standard English?

Reviewer #4: No

Reviewer #5: Yes

Reviewer #6: Yes

6. Review Comments to the Author

Reviewer #4: 1.not following Vancouver reference styling in citation (please refer to submission guidelines)

3.English language used is not professional for article research

4.Abstract should be restructure

Reviewer #5: Good luck

Reviewer #6: 1. Please write the full form acronym (eg PEPFAR) when used for the first time in the text.

2. In the study site and population - please share if written informed consent has been taken from the individuals or not. It is not clear.

3. How was the confidentially of the individuals maintained - should be written in detail.

4. Please REFRAME - Persons accessing ART at the hospital were SELECTED and approached after they had received their service, and the purpose of the study was explained to them. (how can a researcher select without written informed consent and first enrolling individual in the study)

7. PLOS authors have the option to publish the peer review history of their article (what does this mean?). If published, this will include your full peer review and any attached files.

**Do you want your identity to be public for this peer review?** For information about this choice, including consent withdrawal, please see our Privacy Policy.

Reviewer #4: **Yes: **Reema Mohammed Almadodi

Reviewer #5: No

Reviewer #6: **Yes: **Ravleen Kaur Bakshi

---

## [Editor Report · Decision Letter 2]

10 May 2024

Effect of Status Disclosure on Quality of Life among People Living with HIV/AIDS in Ghana: A Health Facility-based Cross-sectional Study

PMEN-D-23-00040R2

Dear Mr. Saah,

We are pleased to inform you that your manuscript 'Effect of Status Disclosure on Quality of Life among People Living with HIV/AIDS in Ghana: A Health Facility-based Cross-sectional Study' has been provisionally accepted for publication in PLOS Mental Health.

Best regards,

Isabella Giulia Franzoi, Ph.D.

Academic Editor

PLOS Mental Health